# Influence of Copper Addition on Sigma Phase Precipitation during Hot Deformation of Duplex Steel

**DOI:** 10.3390/ma13071665

**Published:** 2020-04-03

**Authors:** Grzegorz Stradomski, Arkadiusz Szarek, Dariusz Rydz

**Affiliations:** 1Faculty of Production Engineering and Materials Technology, Czestochowa University of Technology, 19 Armii Krajowej Av. 42-200 Czestochowa, Poland; imitm@wip.pcz.pl; 2Faculty of Mechanical Engineering and Computer Science, Czestochowa University of Technology, 21 Armii Krajowej Av. 42-200 Czestochowa, Poland; itm@itm.pcz.pl

**Keywords:** duplex steel, hot deformation, sigma phase, plasticity, microstructure, influence of copper addition

## Abstract

The paper presents an experimental study on microstructure changes in duplex steel after hot deformation. Duplex steels and cast steels are characterized by a multiphase microstructure. They are relatively new materials with great contributions to the many fields of industries. Due to the fact of deforming two different phase austenite and ferrite those materials have a complex plasticity. This work is a continuation and complementation of previous works and is a significant supplement to information presented in them. The article concerns precipitation phenomena and changes in the microstructure of two grades of ferritic-austenitic steels: X2CrNiMoN25-7-4 and X2CrNiMoCuN25-6-3. Those steels have a very similar chemical composition, differing by only 2.5% copper content. An important aspect presented in the work is we observed that adding 2.5% copper prevented precipitation of the destructive sigma phase during the hot deformation.

## 1. Introduction

Steels and other iron alloys are very well known as construction materials, are widely-used and in the near future this will not change. Among the resistant-to-corrosion steels (Figure 1) and cast steels, the most modern and dynamically developing group are ferritic-austenitic alloys [1,2,3,4,5,6,7,8,9,10,11,12,13,14,15,16,17,18,19]. They are commonly known as duplex. The genesis and development of duplex steels is associated with the appearance of stainless steels in the early twentieth century. In Sheffield (UK), it was found that the addition of chromium at approximately 13% causes electrochemical corrosion resistance [6]. The main application area of duplex steels and cast steels are structures and elements exposed to high loads and environments conducive to stress, pitting or crevice corrosion. For most resistant-to-corrosion steel grades, local corrosion (pitting, crevice, intergranular) is much more dangerous than general corrosion. Hence, modifications and optimization of chemical compositions and microstructures are applied. The development of the special ferritic-austenitic steels group allows us to increase the durability of elements exposed to erosive and corrosive wear or the exploitation of deeply lying, highly sulfurized gas and oil deposits [1,2,3,4,5,7,8,9,10,11,12,13,14,15,16,17,18,19,20,21].

Taking under consideration such conditions, duplex steels and cast steels with a comparable proportion of the basic phases—ferrite and austenite—present a better set of mechanical properties compared to traditionally used ferritic or austenitic steels. The main area of application are the chemical industry, construction of warehouses, tanks of vessels transporting products with high chemical activity (acid phosphoric, concentrated sulfuric acid, highly alkaline media) or equipment used: in the energy production, pulp, paper, food and petrochemical industries [9,10,11,12,13,14,15,16,17]. Figure 2 show the key advantages of duplex steels and cast steels.

## 2. Material and Research Methodology

The subjects of the study were two grades of duplex steels X2CrNiMoN25-7-4 and X2CrNiMoCuN25-6-3, which chemical compositions are presented in Table 1. The microstructure of these materials were shown in earlier works [1,2,3,10]. They are characterized by low tendency to crystalize in form of column grains, which appearance is associated with an increase of carbon content—above 0.04% C for copper-free steel X2CrNiMoN25-7-4 and 0.035% C for X2CrNiMoCuN25-6-3. The chemical composition of materials was selected in such way that the content of key elements was very similar. The main difference was the addition of about 2.5% of copper, which not only allows the aging process [14,22,23,24], but also affects the liquidity and plasticity of duplex steel and cast steel [1,2,3,8,9,10,11,12,13,14,15,16,17,25,26,27].

The conducted experimental researches were divided into the following stages:Making of Y-shaped castings with a wall thickness of 25.0 mm (Figure 3);Analysis of the microstructure of the materials in the raw state;Solution-annealing to remove the primary precipitated sigma phase;Hot deformation using the Gleeble 3800 Physical Simulation System with the Hydrawedge module;Analysis of materials microstructure after the plastic deformation process.

## 3. Obtained Results

The microstructure of analyzed materials was revealed with the metallographic reagent Mi21Fe (30 g potassium ferricyanide, 30 g potassium hydroxide 60 g distilled water). The main advantage of this reagent was marking on different colors phases, what allows and helps easily analysis of microstructure. Figure 4 shows an example of microstructure-tested material in a raw state. As it can be seen the sigma phase have a specific brown color, austenite white or gray and ferrite light brown or yellow. This type of reagent also give the possibility to reveal other phase like the γ’ known also as γ_2_.

The structure of materials were analyzed with use of the light microscope and the phase content was determined using the Nis-Elements D application. The evaluation of the raw state microstructure showed that the material was characterized by presence of ferritic–austenitic grains with a small presence of 2%–2.5% of the σ phase. According to literature study [5,14,18,19,28,29,30] and the authors’ research [2,10,17,31], this phase volume was not harmful.

In the first stage of studies were made deformation of the raw state. In some cases of hot deformed samples, the cracks were observed. This was a basis for the microstructure analysis and some selected results are shown in Figure 5.

Evaluation of the microstructure showed that below 1100 °C, can appear the initiation of phase σ precipitation, what was observed for a deformation rate of 10.0 s^−1^. Quantitative analysis allowed to determine its share at the level of 10%–15% depending on the deformation temperature. The volume fraction of the phase increases with temperature decrease. Based on the obtained data, it was found that, the limit temperature below which the alloy should not be deformed was 1100 °C and it was definitely safer to apply lower deformation rates.

Hot deformation tests were also carried out for the X2CrNiMoCuN25-6-3 alloy, which was presented earlier works [1,3,14]. Cracks were also observed for this alloy using the same parameters as for the X2CrNiMoN25-7-4 alloy, but no new σ phase precipitates were observed. The initial share of this phase was at a similar level of about 2%–2.5%. Therefore, it was proposed to carry out the solution-annealed process to eliminate it completely. The material prepared for deformation was solution-annealed at 1150 °C for 120 min. Figure 6 and Figure 7 show examples of tested-steel microstructures after solution-annealed process.

As it can be seen the material without cooper was characterized by presence of the austenite and ferrite. In the duplex steel with addition of cooper was also present the γ’ also named γ_2_ phase [16,23,25,29,30,32].

In the further part of the work, deformation tests were carried out for both tested alloys according to the parameters presented in Table 2 and Table 3. Temperatures 850 and 1100 °C were selected on the basis of earlier tests, as well as the range of strain rate and the values of the true deformation.

The obtained results (no crack observed) showed that the X2CrNiMoN25-7-4 steel was characterized by good plasticity in the tested temperature range for true deformation 0.2 and 0.3 for both deformation rates 1.0 and 10 s^−1^. Figure 8, Figure 9, Figure 10, Figure 11, Figure 12, Figure 13, Figure 14 and Figure 15 show examples of microstructures deformed according to different variants according to Table 2.

The deformation according to variant 16 led to the crack of the sample. It should be emphasized, that however in this case a total relative deformation of 1.2 was achieved in a relatively short time and with a strain rate of 10 s^−1^.

The true deformation in one pass by 0.6 at a strain rate of 10 s^−1^ at 850 °C (Figure 13) causes the precipitation of the σ phase. They are in the form of single colonies separated from each other by the ferrite–austenite border as well as small in the ferrite, near the borders with austenite. Quantitative analysis showed the presence of about 3%–5% of the newly separated σ phase. Figure 13 show the microstructure of the sample after deformation according to variant 15.

Microstructure after deformation according to variants 13 and 14 are presented in Figure 14 and Figure 15. As a result of the tests, the material subjected to 0.45 true deformation do not crack, while the sample subjected to 0.5 true deformation cracked. It should be noted that, the application of 0.5 true deformation led to the appearance of very small σ phase precipitation. They are located in the area of the largest strains on the ferrite–austenite boundaries (Figure 14). The number of precipitates was negligible and limited only to the zone with the largest deformation.

The use of four passes with true deformation 0.3, strain rate of 10.0 s^−1^ at the temperature of 850 °C also caused the precipitation of the σ phase, which are shown in Figure 16. Despite the fact that the deformation 0.3 was defined as safe, the deformation rate and short break time of 0.25 s probably do not allow for full stress relaxation. In this variant, the location and percentage of new precipitates was similar to that observed in variant 15, with the difference, that more precipitates were visible as elongated colonies at the ferrite–austenite borders.

Based on the tests, was possible to develop the relationship of the percentage share of phase σ as a function of time and strain in the form of an equation and the surface shown in Figure 16.
σ_def_ = −0.004ε^2^ − 5128.426 t_ε_^2^ − 0.314 ε + 156.068 t_ε_ + 12.287ε·t_ε_ + 0.017
where:σ_def_—σ phase percentage,ε—strain, %, t_ε_—deformation time, s.

The X2CrNiMoCuN25-6-3 steel, similarly to the grade without copper, was subjected to solution-annealed process in order to completely eliminate σ phase precipitations. The tests were carried out according to the parameters shown in Table 3. Figure 17, Figure 18, Figure 19, Figure 20, Figure 21 and Figure 22 show the microstructure of the material subjected to deformation according to variants: 1a, 2a, 5a, 13a, 14a and 15a. The deformed sample according to variants 14a and 15a cracks, similarly to sample variant 14, what indicates that the material exceeded the maximum stress values. In the case of these variants it should be emphasized that the test of repeated deformation was dynamic, fast and the time of single deformation was about 0.05 s.

Similar to X2CrNiMoN25-7-4 steel, it has equiaxed ferrite and austenite grains in its raw state. The analysis of microstructure of the X2CrNiMoCuN25-6-3 steel did not show any presence of the σ phase. Although as it was mentioned in some cases were observed cracks, but in the area around them—and in the deformation zone—no precipitation of this phase were observed. In the materials was also noticed no presence of the ε phase, which according to [23,24,32,33] should be observed. As it was known this phase reduce hot ductility and gave the possibility for precipitation hardening. However, some authors [34] who investigated duplex steels with copper content about 2.5% have not found it, but others [35] noticed its presence in materials with about 3% of Cu.

The obtained results clearly shows that the 2.5% copper addition, in the studied temperature–deformation range, eliminates the tendency to precipice the σ phase as a result of hot plastic deformation.

## 4. Summary

The plasticity analysis of X2CrNiMoN25-7-4 steel showed that the delivery state was an important parameter affecting hot plastic deformability. The material deformed in preliminary tests, not subjected to solution-annealed, was characterized by about 2%–2.5% of the σ phase share and lower plasticity. The tests carried out in the temperature range 800–1150 °C showed a tendency to form cracks. I was found that for this deformation rate below 1100 °C, the initiation of phase σ separation may occur, which is shown in Figure 6e,f. Quantitative analysis showed that, depending on the temperature, its share can be 10%–15%. This was a significant increase of 4–6 times. In all areas of cracks of deformed samples in the temperature range of 800–1050 °C for strain rate 10.0 s^−1^ were visible precipitations, which means that it can both initiate and facilitate destruction. It should also be noted that in microstructure of samples deformed at 1100 °C, no σ phase was observed in the areas of cracks.

The materials subjected to solution-annealed at 1150 °C for 120 min had significantly better plastic properties. The obtained results showed that the tested steels were characterized by good plasticity in the tested temperature range for true deformations 0.2 and 0.3 for both strain rates 1.0 and 10 s^−1^. It should be noted that 0.5 deformation according to variant 14 caused the appearance of σ phase precipitates, however, its share was negligible, and the precipitations were very small. This is the proof that this is the initial stage, which should be considered as the moment of the beginning of its separation in the studied temperature and deformation conditions. The deformation in one pass by 0.6 at a speed of 10 s^−1^ at a temperature of 850 °C (Figure 13) causes the separation of the σ phase in the form of single, separated colonies at the ferrite-austenite borders as well as small precipitates in the ferrite, near the borders with austenite. This area was a place of privileged sigma phase release, which was associated with the phenomena of Cr and Ni microsegregation for ferrite and austenite. The use of four passes with strain 0.3 and the strain rate of 10 s^−1^ at temperature of 850 °C also caused the separation of the σ phase. The observed amount was similar to variant 15, with the difference that more precipitates have the character of longitudinal colonies at the ferrite–austenite boundaries. In this case, despite the fact that the actual strain 0.3 was defined as safe, the strain rate and short break time probably do not allow to full stress relaxation.

The plasticity of the X2CrNiMoCuN25-6-3 steel was similar to the X2CrNiMoN25-7-4 steel. Although both materials cracks during performed tests. In the microstructure of the grade with copper was not observed σ phase. This steel was characterized by the presence of not only main phases like austenite and ferrite but also the γ’ (γ2). The only observed precipitations of sigma phase were observed during initial tests which concerned no solution-annealed material. In those tests the amount of precipitated σ phase during crystallization was preserved and after deformation its amount was similar. After solution-annealing process in the microstructure of the steel with 2.5% of copper, deformed in the 850 °C, none σ phase precipitation were observed. It should be mentioned that deformation in this range of temperature can provide the grain refining and changes in texture as it is presented in [19]. Lack of σ phase precipitation give the possibility for deformation, for example rolling process in the range of 600–900 °C. This is the temperature of creation of σ phase.

## Figures and Tables

**Figure 1 materials-13-01665-f001:**
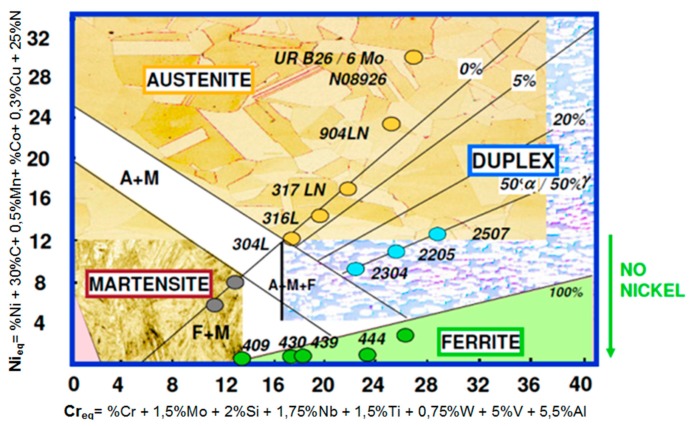
Schaeffler’s graph—Based on [4,5,7,8,9,14,16].

**Figure 2 materials-13-01665-f002:**
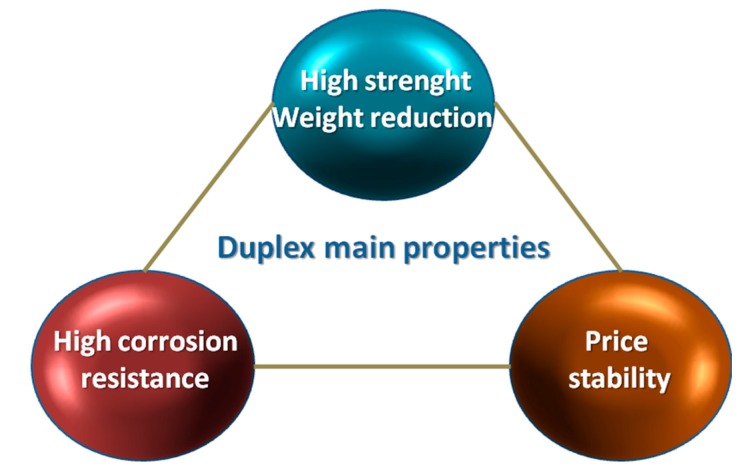
Schematic presentation of the most important advantages of duplex steels and cast steels. Based on [15].

**Figure 3 materials-13-01665-f003:**
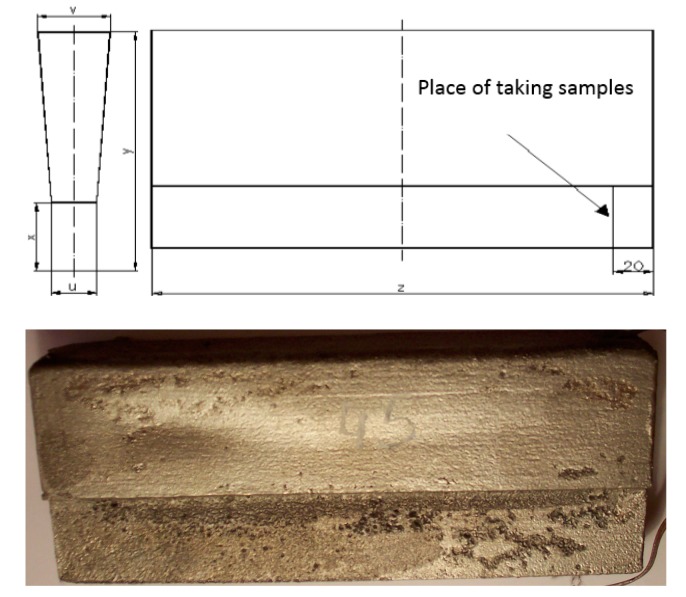
The shape of the cast with head.

**Figure 4 materials-13-01665-f004:**
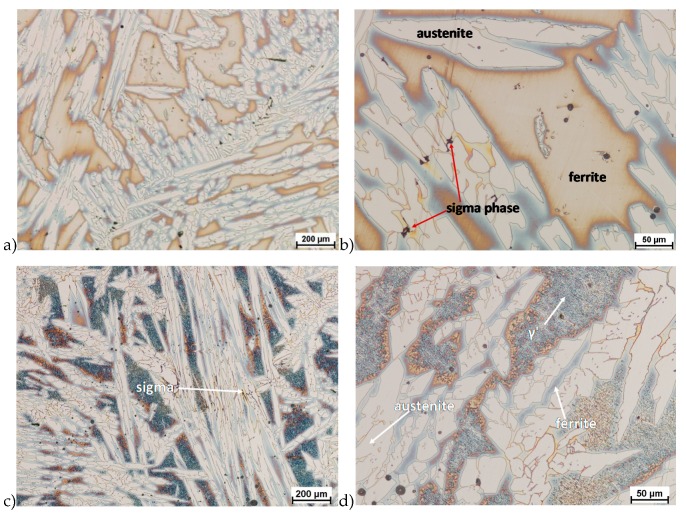
Exemplary microstructure in the raw state (**a**,**b**) the X2CrNiMoN25-7-4 steel, (**c**,**d**) the X2CrNiMoCuN25-6-3 steel.

**Figure 5 materials-13-01665-f005:**
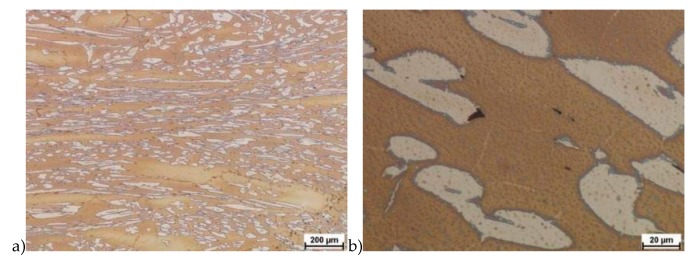
Exemplary microstructures of the X2CrNiMoN25-7-4 steel: **a**) deformation at temperature 1150 °C with strain rate 1.0 s^−1^, **b**) deformation at temperature 1150 °C with strain rate 1.0 s^−1^, **c**) deformation at temperature 1100 °C with strain rate 10.0 s^−1^, **d**) deformation at temperature 1100 °C with strain rate 10.0 s^−1^, **e**) deformation at temperature 900 °C with strain rate 10.0 s^−1^, **f**) deformation at temperature 900 °C with strain rate 10.0 s^−1^.

**Figure 6 materials-13-01665-f006:**
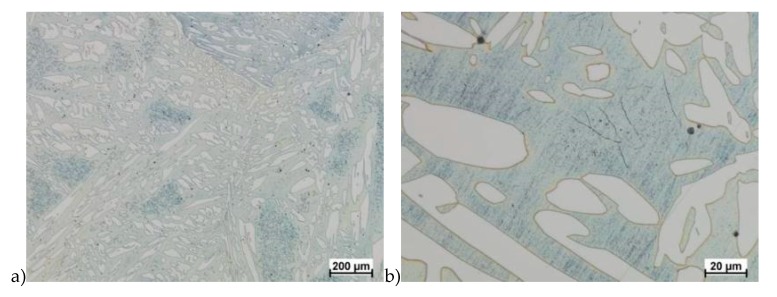
Microstructure of the X2CrNiMoN25-7-4 steel before deformation. **a**) magn. 50×, **b**) 500×.

**Figure 7 materials-13-01665-f007:**
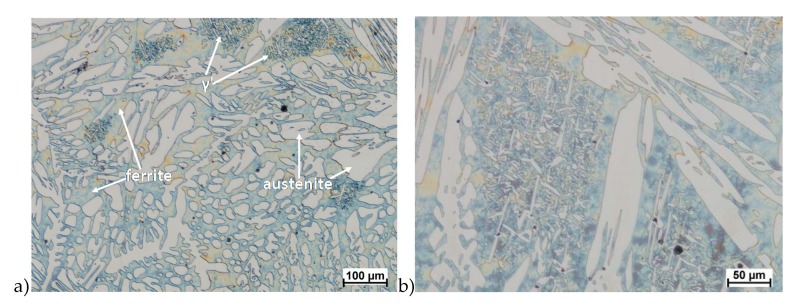
Microstructure of the X2CrNiMoCuN25-6-3 steel before deformation. **a**) magn. 100×, **b**) 200×.

**Figure 8 materials-13-01665-f008:**
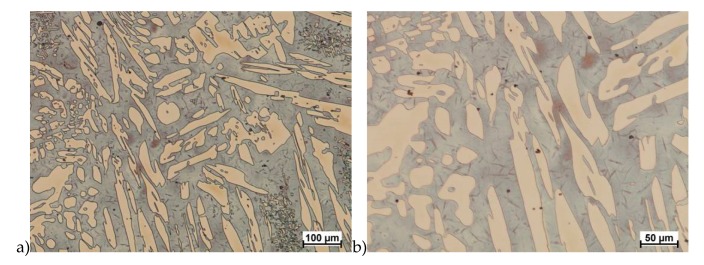
Microstructure of the X2CrNiMoN25-7-4 steel deformed according to the variant 1. **a**) magn. 100×, **b**) 200×.

**Figure 9 materials-13-01665-f009:**
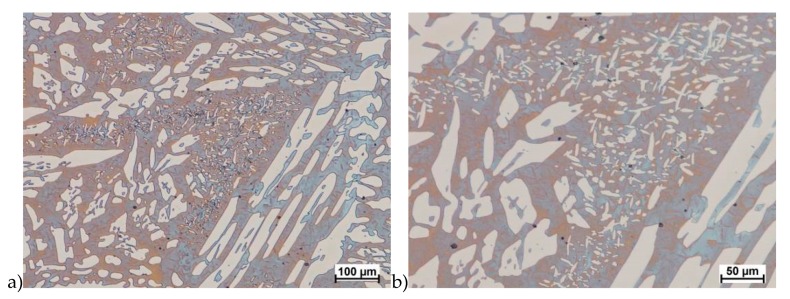
Microstructure of the X2CrNiMoN25-7-4 steel deformed according to the variant 2. **a**) magn. 100×, **b**) 200×.

**Figure 10 materials-13-01665-f010:**
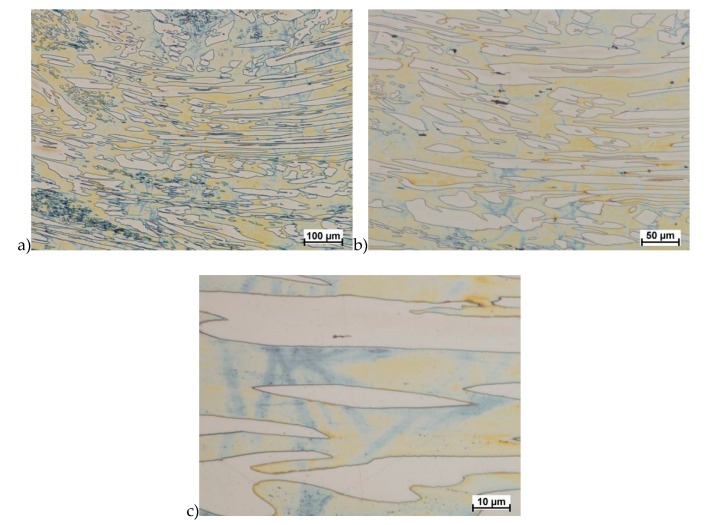
Microstructure of the X2CrNiMoN25-7-4 steel deformed according to the variant 5. **a**) magn. 100×, **b**) 200×, **c**) 1000×.

**Figure 11 materials-13-01665-f011:**
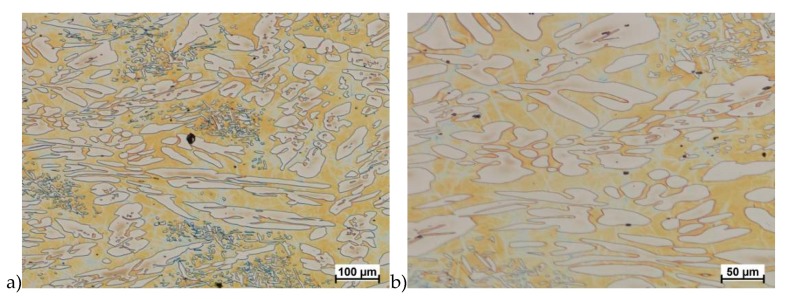
Microstructure of the X2CrNiMoN25-7-4 steel deformed according to the variant 8. **a**) magn. 100×, **b**) 200×.

**Figure 12 materials-13-01665-f012:**
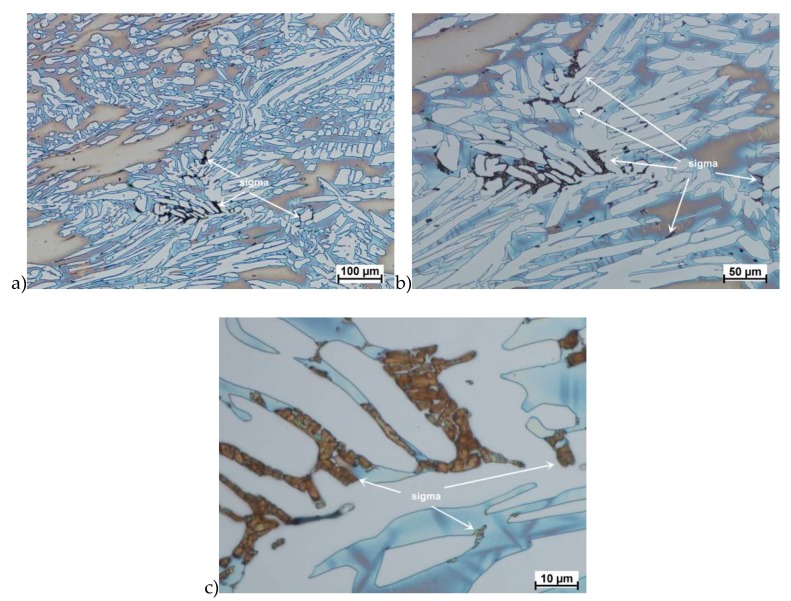
Microstructure of the X2CrNiMoN25-7-4 steel deformed according to the variant 15. **a**) magn. 100×, **b**) 200×, **c**) 1000×.

**Figure 13 materials-13-01665-f013:**
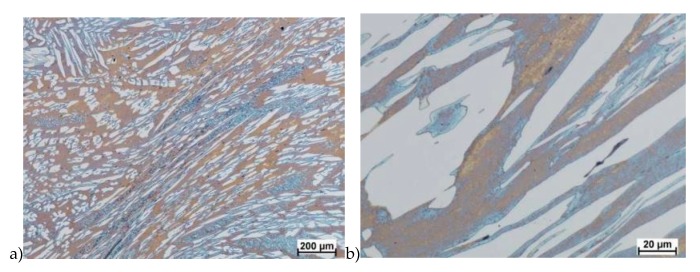
Microstructure of the X2CrNiMoN25-7-4 steel deformed according to the variant 13. **a**) magn. 50×, **b**) 500×.

**Figure 14 materials-13-01665-f014:**
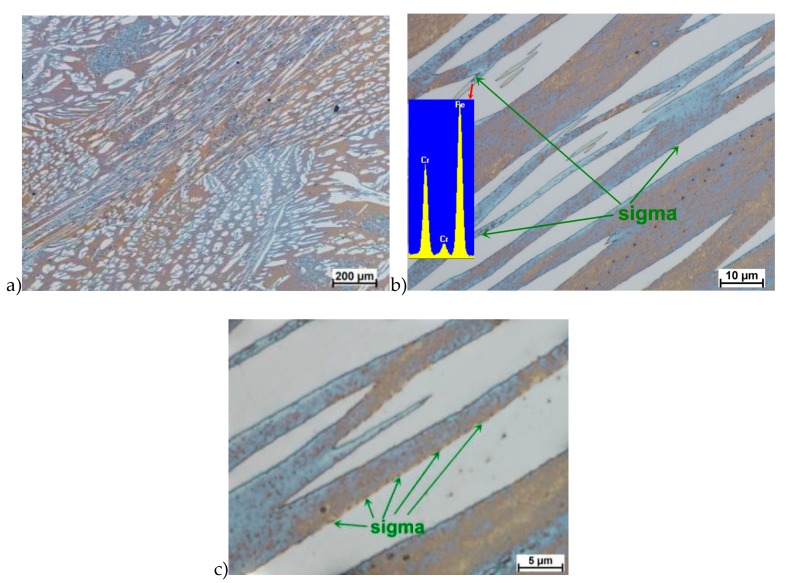
Microstructure of the X2CrNiMoN25-7-4 steel deformed according to the variant 14. **a**) magn. 50×, **b**) 1000×, **c**) 2250×.

**Figure 15 materials-13-01665-f015:**
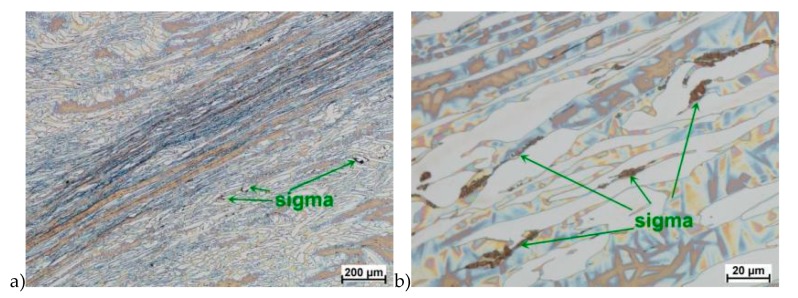
Microstructure of the X2CrNiMoN25-7-4 steel deformed according to the variant 16. **a**) magn. 50×, **b**) 500×.

**Figure 16 materials-13-01665-f016:**
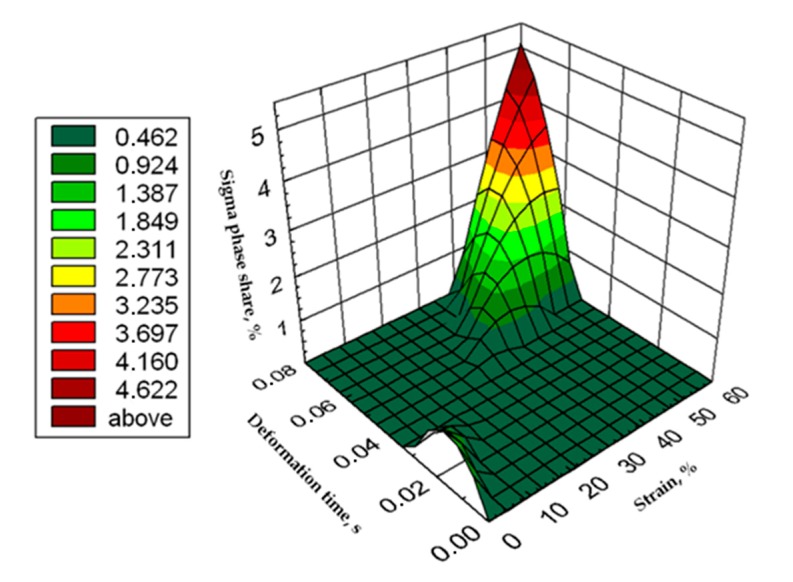
Graphic presentation of the relationship of the percentage share of σ phase as a function of time and strain for X2CrNiMoN25-7-4 steel at 850 °C.

**Figure 17 materials-13-01665-f017:**
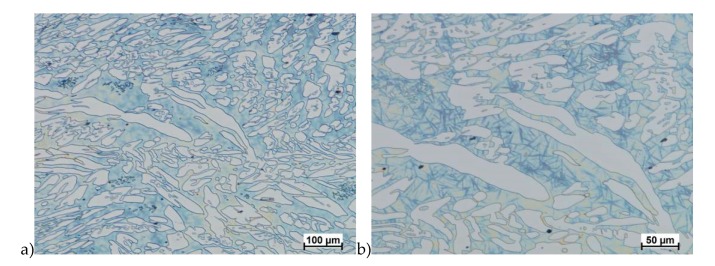
Microstructure of the X2CrNiMoCuN25-6-3 steel deformed according to the variant 1a. **a**) magn. 100×, **b**) 200×.

**Figure 18 materials-13-01665-f018:**
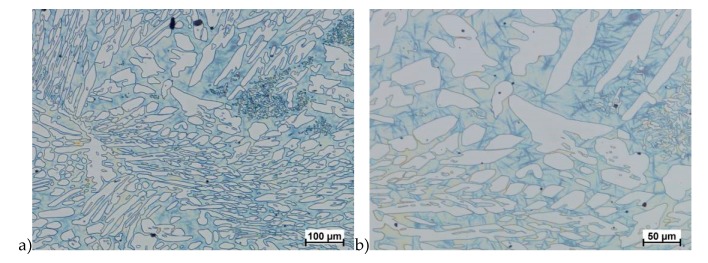
Microstructure of the X2CrNiMoCuN25-6-3 steel deformed according to the variant 2a. **a**) magn. 100×, **b**) 200×.

**Figure 19 materials-13-01665-f019:**
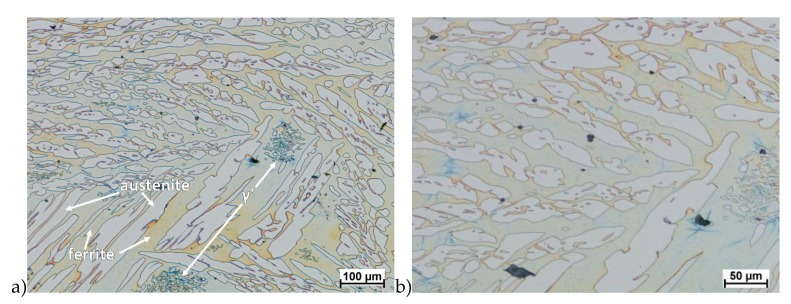
Microstructure of the X2CrNiMoCuN25-6-3 steel deformed according to the variant 5a. **a**) magn. 100×, **b**) 200×.

**Figure 20 materials-13-01665-f020:**
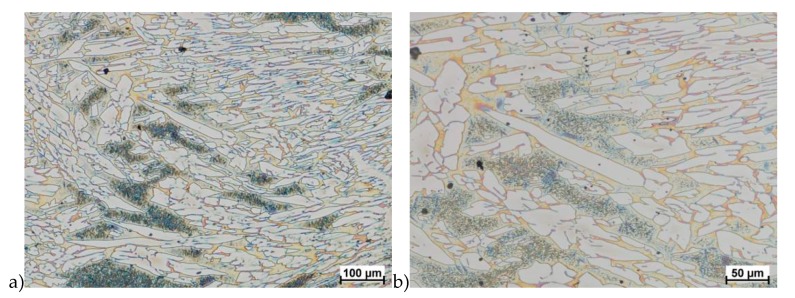
Microstructure of the X2CrNiMoCuN25-6-3 deformed according to the variant 13a. **a**) magn. 100×, **b**) 200×.

**Figure 21 materials-13-01665-f021:**
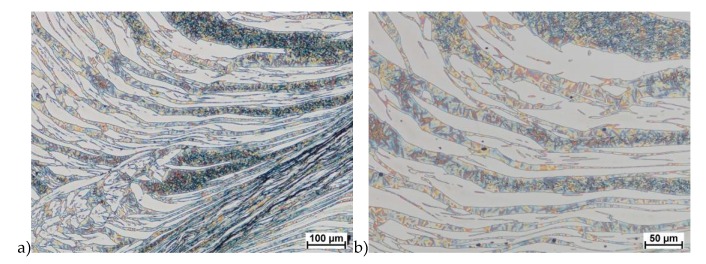
Microstructure of the X2CrNiMoCuN25-6-3 deformed according to the variant 14a. **a**) magn. 100×, **b**) 200×.

**Figure 22 materials-13-01665-f022:**
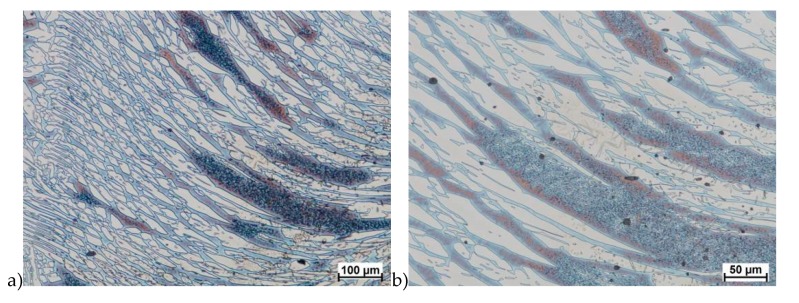
Microstructure of the X2CrNiMoCuN25-6-3 deformed according to the variant 15a. **a**) magn. 100×, **b**) 200×.

**Table 1 materials-13-01665-t001:** Chemical composition of tested steels according to standard PN-EN 10283: 2019, value in weight %.

Designation	C	Mn	Si	S	P	Cr	Ni	Mo	Cu	N
X2CrNiMoN25-7-4	0.021	1.46	0.93	0.012	0.008	26.70	6.48	3.10	0.02	0.23
	according to the standard									
	0.03 *	2.0 *	1.0 *	0.015 *	0.035 *	24.0–26.0	6.0–8.0	3.0–4.5	-	0.2–0.35
X2CrNiMoCuN25-6-3	0.024	1.32	0.81	0.011	0.008	25.84	6.34	2.93	2.75	0.23
	according to the standard									
	0.03 *	2.0 *	0.7 *	0.015 *	0.035 *	24.0–26.0	5.0–7.5	2.7–4.0	1.0–2.5	0.15–0.35

* maximum values specified in the standard.

**Table 2 materials-13-01665-t002:** Temperature and deformation parameters used during tests for X2CrNiMoN25-7-4 steel.

Design	Temperature, °C	Strain Rate, s^−1^	True Deformation	Pause Time, s
1	850	1.0	ε1 = 0.2	-
2	850	10.0	ε1 = 0.2	-
3	1100	1.0	ε1 = 0.2	-
4	1100	10.0	ε1 = 0.2	-
5	850	1.0	ε1 = ε2 = ε3 = 0.2	0.25
6	1100	1.0	ε1 = ε2 = ε3 = 0.2	0.25
7	850	1.0	ε1 = 0.3	-
8	850	10.0	ε1 = 0.3	-
9	1100	1.0	ε1 = 0.3	-
10	1100	10.0	ε1 = 0.3	-
11	850	1.0	ε1 = ε2 = ε3 = 0.2	0.25
12	850	10.0	ε1 = 0.4	-
13	850	10.0	ε1 = 0.45	-
14	850	10.0	ε1 = 0.5	-
15	850	10.0	ε1 = 0.6	-
16	850	10.0	ε1 = ε2 = ε3 = ε4 = 0.3	0.25

**Table 3 materials-13-01665-t003:** Temperature and deformation parameters used during tests for X2CrNiMoCuN25-6-3 steel.

Design.	Temperature, °C	Strain Rate, s^−1^	True Deformation	Pause Time, s
1a	850	1.0	ε1 = 0.2	-
2a	850	10.0	ε1 = 0.2	-
3a	1100	1.0	ε1 = 0.2	-
4a	1100	10.0	ε1 = 0.2	-
5a	850	1.0	ε1 = ε2 = ε3 =0.2	0.25
6a	1100	1.0	ε1 = ε2 = ε3 = 0.2	0.25
7a	850	1.0	ε1 = 0.3	-
8a	850	10.0	ε1 = 0.3	-
9a	1100	1.0	ε1 = 0.3	-
10a	1100	10.0	ε1 = 0.3	-
11a	850	1.0	ε1 = ε2 = ε3 =0.2	0.25
12a	850	10.0	ε1 = 0.4	-
13a	850	10.0	ε1 = 0.6	-
14a	850	10.0	ε1 = ε2 = ε3 = 0.3	0.25
15a	850	10.0	ε1 = ε2 = ε3 = ε4 = 0.3	0.25

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
