# Peer review of "Influence of Copper Addition on Sigma Phase Precipitation during Hot Deformation of Duplex Steel"

_materials, 2020, doi:10.3390/ma13071665_

Round 1

Reviewer 1 Report

Dear authors,

Please take into consideration the following:

The observations stated in this paper are quite interesting and odd. Robert Gunn is the book titled as ‘Duplex Stainless Steels’ on page 19 states that: “In general, the addition of Cu to duplex stainless steels is limited to about 2 %, since higher levels reduce hot ductility and can lead to precipitation hardening (see Section 3.4.8).”

Is there any constitutional diagrams, which take copper into attention as an austenite former?

In case of Figure 1. the nickel equivalent is not visible and the chromium equivalent is modified compared to the original by Schaeffler. The authors should also refer to the original diagram in the caption if this figure is really needed in this context.

In my opinion, Figure 2 is unnecessary.

There are some editorial problems, beside the English language should be checked carefully, such as page 2, lines 45-49.

In my opinion, Figure 3. is unnecessary.

Although it is evident for duplex stainless steel researchers, I would recommend to show austenite and ferrite phases in the first microstructural image (Figure 5).

In figure captions the magnification is unnecessary.

In Figure 5. some precipitations are visible? If yes, what are those?

Instead of ‘supersaturated’ I would use the term ‘solution-annealed’.

Tables 2 and 3 should be formatted to have an easier overview for the reader.

Figures 9-11 showing some precipitations like areas within the ferrite phase. Did the authors further investigate these phases?

Did the authors quantify the amount the sigma phase in the case of the 850 °C deformed sample? If yes, a summary table would be useful.

In Figure 15 the sigma phase is hardly visible. I would recommend to use other etchants, which were designed to reveal sigma phase in the microstructure. A collection of different etchants for different purposes can be found for example in https://doi.org/10.3311/PPme.12234.

Did the authors find any presence of epsilon phase?

Author Response

Thank you for the valuable comments that have been taken into account. Detailed explanations are in the attachment.

Reviewer 2 Report

The manuscript of Grzegorz Stradomski stats that will present the influence of copper addition on sigma phase precipitation during hot deformation of a duplex steel. After reading the paper the conclusion is that the authors fail to present this influence. The manuscript is difficult to be read and is hard to see where is presented a new work and where is a presentation of an already published work. The references are not enough in number, half of them are the authors own papers, some of them in Polish language.

In the introduction the authors could include their own work, as basis of a further study presented below.

Table 2 and 3 are identically, presentation of only one of them is enough.

Page 7, line 140: “The obtained results showed that the X2CrNiMoN25-7-4 steel is characterized by high plasticity” where are these results?

The microstructures of the used conditions are not present consistent for both materials.

The conclusions are not sufficiently concise.

Author Response

(The authors gave the same response as above.)

Round 2

Reviewer 2 Report

The authors have imprved their paper and deserve to be published.